# What Is the Impact of COVID-19 Pandemic on Patients with Pre-Existing Mood or Anxiety Disorder? An Observational Prospective Study

**DOI:** 10.3390/medicina57040304

**Published:** 2021-03-24

**Authors:** Antonio Tundo, Sophia Betro’, Roberta Necci

**Affiliations:** Istituto di Psicopatologia, 00196 Rome, Italy; sophia.bet@gmail.com (S.B.); necci@istitutodipsicopatologia.it (R.N.)

**Keywords:** coronavirus, panic disorder, social anxiety disorder, major depressive disorder, bipolar disorder, obsessive–compulsive disorder, psychological impact

## Abstract

*Background and Objectives:* This observational prospective study aims to examine the psychological and psychopathological impact of the pandemic stress on patients with pre-existing mood, anxiety and obsessive–compulsive disorders. *Materials and Methods:* The study includes 386 consecutive patients recruited from 10 March to 30 June 2020 among those being treated at the Institute of Psychopathology in Rome (Italy) with an age ≥18 years and meeting DSM-5 criteria for major depressive disorder (MDD) (35.2%), bipolar I (BD-I) (21.5%) or II (BD-II) (28.8%) disorder, obsessive–compulsive disorder (OCD) (7.5%), panic disorder (PD) (7.0%) or social anxiety (SA). A total of 34.2% had lifetime comorbid Axis I disorders and 15.3% had alcohol/drug abuse disorders. Using a semi-structured interview, we investigated if the impact of COVID-19 stress for patients has been similar, higher or lower than that of their family and friends and, for patients with relapse/symptoms worsening, if there was a relationship between the clinical condition worsening and the pandemic stress. *Results:* Compared with that experienced by their family members and friends, the psychological impact of pandemic stress was similar in 52.1% of the sample, better in 37.1% and worse in 10.8%. In 21 patients (5.4%), the stress triggered a recurrence or worsened the symptoms. Patients with OCD had a higher rate of worsening due to pandemic stress compared to patients with MDD (*p* = 0.033), although, overall, the χ^2^ test was not significant among primary diagnoses (χ^2^ = 8.368; *p* = 0.057). *Conclusions:* The psychological and psychopathological consequences of COVID-19 stress in our outpatients were very modest. The continuity of care offered during the lockdown could explain the results.

## 1. Introduction

Coronavirus disease (COVID-19), declared a pandemic on 11 March 2020 by the World Health Organization [1], rapidly spread from China to the world and changed the lifestyle of a large number of people. After China, Italy was the second large country infected and adopted a strict lockdown extended to the entire population. The COVID-19 pandemic has the potential to cause severe mental health problems both directly (fear of becoming ill and dying) and indirectly (social isolation related to quarantine, financial burden, public transportation restrictions, school closure). Several studies, mainly conducted through internet surveys or questionnaire administration, reported a 20% increase in anxiety and depression symptoms [2,3] and a 32% increase in alcohol use [4,5] in the general population. The increase in anxiety symptoms, in turn, could explain the increase in benzodiazepine consumption in Italy [6] as well as in other countries [7,8]. For example, the Italian Medicines Agency (AIFA) reported that the mean number of benzodiazepine packs for 10,000 residents purchased by pharmacies from March to May 2020 was significantly higher than that purchased from December 2019 to February 2020 (24.11 and 23.22, respectively; *p* = 0.000) [6].

Several authors suggested that people with a previous history of mood disorder (MD) or anxiety disorder (AD) are at high risk of symptom worsening during the COVID-19 pandemic because they are more vulnerable than the general population to the fear of getting sick and to the changes in lifestyle related to quarantine [9,10,11,12,13]. However, the observational data on the topic are scant and highly controversial. A worldwide survey reported the worsening of psychiatric conditions in two thirds of 2734 psychiatric patients, with an increase in the scores on scales for psychological disturbance, posttraumatic stress disorder and depression [14]. Four clinical studies found a worsening of psychiatric conditions in 20–50% of patients with pre-existing anxiety disorders, depressive disorders and obsessive–compulsive disorder (OCD), and in 16% of patients with substance use disorders [15,16,17,18]. Three other clinical studies found opposite results. The impact of pandemic stress triggered only a minimal increase in symptomatology or social impairment in a sample of 54 patients with anxiety disorders [19], no changes in mood or sleep duration in 56 patients with affective disorders [20] and no increase in depression, anxiety and suicidal ideation in 73 old patients with major depressive disorder [21].

Since the COVID-19 pandemic is still expanding in the word, further information on its impact on people with psychiatric disorders, considered most vulnerable to the direct and indirect consequences of the pandemic, is essential for clinicians and for mental health care professionals.

The present study aims to examine in a clinical setting the psychological and psychopathological impact of COVID-19 stress on outpatients with pre-existing MD, AD or OCD.

## 2. Materials and Methods

### 2.1. Participants

This observational prospective study included a cohort of patients consecutively recruited from 10 March (start of lockdown in Italy) to 30 June (1.5 months after the stop of lockdown) 2020 at the Institute of Psychopathology in Rome, Italy, an Italian private center specialized in mood and anxiety disorders. Inclusion criteria were: (1) age ≥18 years; (2) meeting DSM-5 criteria [22] for major depressive disorder (MDD), bipolar I (BD-I) or II (BD-II) disorder, panic disorder (PD), obsessive–compulsive disorder (OCD) and social anxiety (SA); (3) being treated at the Institute before March 2020. The presence of mood/anxiety disorder comorbidity or of alcohol abuse/drug use disorder lifetime comorbidities was not an exclusion criterion. Written informed consent for the anonymous use of clinical records was collected routinely at patients’ first visit. The procedure was approved by the local ethical committee (Roma 30 July 2019; Prot. N 1521/CE Lazio 1) and is in accordance with the Helsinki Declaration of 1975, as revised in 2008.

### 2.2. Assessments

All patients were diagnosed using the Structured Clinical Interview for DSM-5 (SCID-5) [23], and at each visit, they were clinically assessed and treated by the first author (AT). The assessment included also the administration of scales to rate the severity of the disorder (Hamilton Depression Rating Scale_21_ [24] and Y-MANIA Rating Scales [25] for MD; Yale-Brown Obsessive Compulsive Scale [26] for OCD; Panic Attack and Anticipatory Anxiety Scale [27] for PD; Brief Social Phobia Scale [28] for SA. The rating scales were administered by the second author (SB), a psychiatrist not involved in the treatment and experienced in mood and anxiety disorders. In our routine practice, we investigated the presence of stressors between visits and their influence on the psychopathological condition. To evaluate the impact of pandemic stress, we used a semi-structured interview in which we asked: (a) if the fear of being infected was not at all/a little bit or moderately/quite a bit distressing; (b) if this fear was higher, similar or lower in terms of distress for him/her than for his/her family and friends; (c) if changes in lifestyle, mostly social isolation, related to quarantine were not at all/a little bit or moderately/quite a bit distressing; (d) if these changes were higher, similar or lower in terms of distress for him/her than for his/her family and friends; (e) if the financial burden due to quarantine was not at all/a little bit or moderately/quite a bit distressing; (f) if this burden was higher, similar or lower in terms of distress for him/her than for his/her family and friends.

Furthermore, we asked patients with relapse/symptoms worsening: (g) if there was a relationship between the worsening of the clinical condition and the stress directly or indirectly related to pandemic stress. As usual in our institute practice, we systematically recorded the number and the content of calls received by our telephone service for psychiatric emergencies.

The first author chose the treatment according to his own clinical experience and the international guidelines for the treatment of MD [29], BD [30], OCD [31], PD and SA [32].

Eighty percent of the visits were conducted online, with a similar duration of face-to-face consultations, and 20% in person, following the government’s safety protocol, from March 10 to May 10, and vice versa subsequently. 

We split the sample into two subgroups: patients reporting relapse/symptom worsening related to pandemic stress and patients who did not. Relapse was defined as follows: no DSM-5 criteria for a disorder and rating scale(s) score below the cut-off at study entry; DSM-5 criteria for at least one disorder and rating scale(s) score over the cut-off during the study period. Symptoms worsening was defined as follows: DSM-5 criteria for at least one disorder during the study, rating scale(s) score over the cut-off at study entry and increase in scores during the study period.

### 2.3. Statistical Analysis 

Categorical variables were summarized using absolute and relative frequencies, and quantitative variables were summarized using mean and standard deviation or median and interquartile range (IQR), according to the frequency distribution of variables.

The association between the categorical variables and the outcome was investigated using χ^2^ or Fisher’s exact test. The Mann–Whitney test was used to compare age and the number of calls between groups.

Statistical analyses were conducted using statistical software IBM SPSS version 25. All tests were two-tailed, and the significance level was set at *p* < 0.05.

## 3. Results

### 3.1. Study Sample Characteristics

The study sample included 386 patients, and 229 (59.3%) were females; mean age was 52.0 ± 16.8 years (range 18–90). During the quarantine, 88 (22.7%) patients were living alone. One hundred and thirty-six patients (35.2%) had a diagnosis of MDD, 111 (28.8%) of BD-II, 83 (21.5%) of BD-I, 29 (7.5%) of OCD and 27 (7%) of PD. Furthermore, 132 patients (34.2%) had at least one lifetime comorbid Axis I disorder (75 (19.4%) OCD, 51 (13.2%) PD, 6 (1.6%) SA) and 59 (15.3%) an alcohol/drug abuse lifetime comorbid disorder.

### 3.2. Psychological Impact of Pandemic Stress

Forty-two of 368 (10.8%) patients reported they experienced a higher distress than their family and friends related to the fear of being infected (18 patients), the changes in lifestyle related to quarantine (21 patients) and the financial burden (3 patients). During the observation period, 21 of 386 (5.4%) patients relapsed/worsened due to COVID-19 distress, while 347 patients did not report any clinical consequence of the pandemic stress. 

Two hundred and one patients (52.1%) reported that their direct (fear of becoming ill) and indirect reactions (social isolation related to quarantine and financial burden) to pandemic stress were similar to those of their family members and friends. One hundred and forty-three patients (37.1%) reported a better adaptation to quarantine than their family and friends. They considered the quarantine an opportunity to spend their time with their family and to have more free time.

### 3.3. Correlates of Pandemic Stress

As shown in Table 1, there were no significant differences between patients with and without relapse/symptom worsening related to pandemic stress regarding sex, age, living alone during the quarantine or Axis I or alcohol/drug abuse lifetime comorbidity. Patients with OCD had a higher rate of worsening due to pandemic stress compared to patients with MDD (13.8% vs. 2.9%, *p* = 0.033), although overall the χ^2^ test was not significant among primary diagnoses (χ^2^ = 8.368; *p* = 0.057).

### 3.4. Calls to the Emergency Service

The number of phone calls for psychiatric emergencies received from 10 March to 30 June 2020 did not differ significantly from that received from 10 March to 30 June 2019 (2177 and 2029, respectively; Mann–Whitney test = 9; *p* = 0.77) (Figure 1). The number of phone calls in March–April 2020 (1080) was higher than that of March–April 2019 (856), and in May–June 2020 (1097), it was slightly lower than that of May–June 2019 (1173). Content related to COVID-19 distress (fear of the infection, increase in washing compulsion, depressive symptoms related to quarantine limitation) in phone calls was recorded for 65 out 2177 calls (2.9%) almost only in March–April.

### 3.5. Treatment Changes

During the observation period, 124 patients (32%) had no change in the treatment schedule and 262 patients (68%) had a change in antidepressant and/or mood stabilizer and/or second-generation antipsychotic treatment. A total of 13/262 patients (3%) received supplementary drugs, mostly benzodiazepines, to reduce anxiety or sleep disorders due to pandemic stress. The 13 patients receiving supplementary drugs to manage the stress were in the subgroup reporting that the pandemic stress negatively influenced their clinical condition, triggering a recurrence or symptoms worsening. 

## 4. Discussion

To our knowledge, this is the first study to evaluate the psychological and the psychopathological impact of stress directly and indirectly related to the COVID-19 pandemic on outpatients with pre-existing MD, AD or OCD recruited and assessed in a clinical setting. 

The findings of the present study indicate good psychological reactions and adaption and a low psychopathological impact of the COVID-19 pandemic on our patients. 

In 50% of patients, the psychological reaction was similar to that of their close friends and relatives without mental disorders and, notably, one patient out of three endorsed the positive aspects of the quarantine, showing a great resilience. Only few patients (11%) showed higher concerns, mainly the fear of becoming ill and the changes in lifestyle.

The clinical consequences of pandemic and quarantine stress in our sample were very modest and only a limited number (less than 6%) of patients reported the emergence of a new episode or the worsening of the symptoms of a pre-existing episode due to COVID-19 distress. OCD, compared to MDD, is the only predictor of increased risk of relapse/symptoms worsening. The exacerbation of OCD symptoms has been reported in a previous study [17], although with a higher rate than in our study (35.8% vs. 13.8%), and could be related to the higher sensitivity of OCD patients to the potential contamination and to the increase in free time during the lockdown, leading to an increase in compulsive behaviors. The sample size for OCD and PD was too small to draw conclusions concerning these diagnoses.

The low psychological and psychopathological impact of pandemic stress reported by our patients is confirmed by the calls to our emergency telephone number. Overall, the number of contacts was quite similar to that of the previous year and the request for help concerning psychological or clinical problems related to COVID-19 stress was limited to 3% of calls and occurred almost only in the first two months corresponding to the peak of the infection and to the more alarming information reported by the media. The high number of phone calls in April–May 2020, corresponding to the Italian complete lockdown, could be due to the fear of leaving the house and the lack of confidence with the technology of some patients, who preferred to call instead of scheduling a follow-up visit online. The low percentage (3%) of patients receiving supplementary drugs to manage the pandemic stress indirectly confirms the modest psychological and psychopathological impact of COVID-19 distress in our sample.

As reported in the Introduction, data on the topic are scant and controversial. Our findings are consistent with those of some previous studies, showing resilience and no worsening of pre-existing mood or anxiety disorder symptoms during the COVID-19 pandemic [19,20,21], but in contrast with those of other studies, reporting a high psychological vulnerability and a worsening of symptoms as a consequence of the pandemic stress [14,15,16,17,18]. 

One possible reason for the conflicting results could be the different methodology in patients’ recruitment. In fact, three of five clinical studies showing a worsening of symptoms were conducted through surveys [14,15,16], while three of four studies not showing a worsening of symptoms were conducted in clinical settings, all in person (including the present study) or partially in person and partially by phone call [19,20].

A second explanation for the conflicting results is the continuity of care. During the pandemic, our patients accessed the website and the administrative service for information and received care as usual (psychiatric visits online or in person and telephone availability for emergency). In the same way, patients continued to receive the treatment in two of three previous studies not showing a worsening of symptoms [19,21]. On the contrary, one of the previous studies showing a worsening of symptoms reported that some patients self-reduced or stopped treatment [15], and the others did not specifically clarify this point but were conducted in countries where, at the time, routine psychiatric counselling was delivered [16,17].

The main limitation of the present study is the imbalance in the outcome ratio. In fact, the ratio between patients experiencing relapse/symptom worsening related to pandemic stress and those who did not experience any relapse/symptom worsening related to pandemic distress is only 0.06. The low statistical power may have limited our ability to detect correlates of relapse/symptom worsening, mostly for the OCD and PD sub-sample. A further limitation is the absence of a control group of untreated patients.

The strengths of this study are that it is based on a real-world clinical sample, and not on a survey, and that it includes self-report and interview-based measures to evaluate the subjective impact of the pandemic and symptom changes.

## 5. Conclusions

In conclusion, our results do not confirm the high vulnerability to the direct and indirect consequences of COVID-19 pandemic stress of patients with pre-existing AD, OCD or MD. The findings of other studies, showing an increase in anxiety and depressive symptoms and alcohol/drug abuse, could be related not to the clinical conditions per se, as currently believed, but to the unavailability of the routine visits with clinicians to change ineffective treatments or address adverse events. The continuity of care can reduce the risk of symptoms exacerbation also in patients with OCD who are more sensitive to pandemic stress, as highlighted by the lower rate of relapse/symptoms worsening in our sample compared to that reported in another study.

Since the COVID-19 pandemic is still expanding in the world, it is urgent that mental health professionals ensure the continuity of care to their patients, adopting a blended approach that combines telehealth and in-person consultations. The continuity of care could mitigate the psychological and psychopathological impact of pandemic stress on patients with mental disorders and prevent recurrences. Further research in clinical settings is warranted to elucidate the protective role of the continuity of care vs. discontinuous care related to difficulties in accessing mental health services during the pandemic.

## Figures and Tables

**Figure 1 medicina-57-00304-f001:**
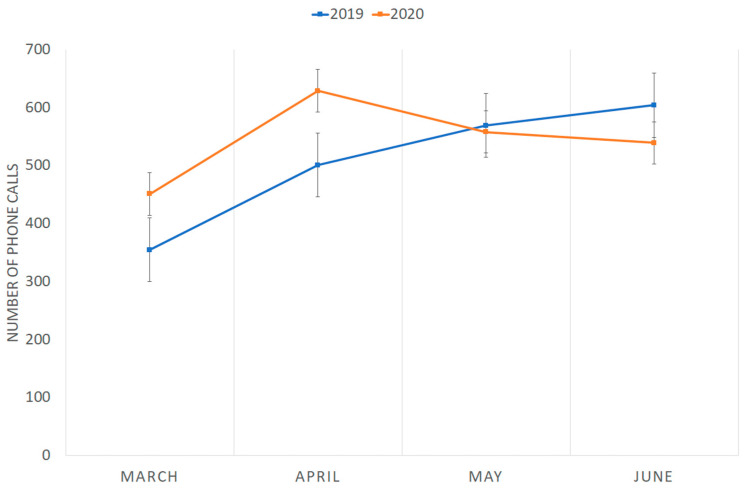
Phone calls to the emergency number received in March–June 2019 and March–June 2020.

**Table 1 medicina-57-00304-t001:** Association of sociodemographic and clinical characteristics with the outcome relapse/worsening (R/W) vs. no relapse/worsening (No R/W) related to COVID-19 distress.

	No R/W (*n* = 365)		R/W (*n* = 21)		Test	*p*-Value
Sex					0.496 *	0.481
Male, *n* (%)	150	41.1%	7	33.3%		
Female, *n* (%)	215	58.9%	14	66.7%		
Age, median (IQR)	52 (40; 65)		48 (43; 57)		168.0 ^#^	0.931
Living alone					0.013 *	0.91
No, *n* (%)	282	77.3%	16	76.2%		
Yes, *n* (%)	83	22.7%	5	23.8%		
Primary diagnosis					8.368 *	0.057
MDD, *n* (row%)	132	97.1%	4	2.9%		
BD-I, *n* (row%)	77	92.8%	6	7.2%		
BD-II, *n* (row%)	107	96.4%	4	3.6%		
OCD, *n* (row%)	25	86.2%	4	13.8%		
PD, *n* (row%)	24	88.9%	3	11.1%		
Comorbidity					6.591 *	0.174
No, *n* (%)	242	66.3%	12	57.1%		
OCD, *n* (%)	68	18.7%	7	33.3%		
PD, *n* (%)	50	13.7%	1	4.8%		
SA, *n* (%)	5	1.3%	1	4.8%		
Abuse					0.243 *	0.622
No, *n* (%)	310	85.0%	17	81.0%		
Yes, *n* (%)	55	15.0%	4	19.0%		

* χ^2^ test. ^#^ Mann–Whitney test. Abbreviations: MDD = major depressive disorder. BD-I = bipolar I disorder. BD-II = bipolar I disorder. OCD = obsessive–compulsive disorder. PD = panic disorder. SA = social anxiety.

## Data Availability

The data presented in the study are available on request from the corresponding author. The data are not publicly available due to privacy restrictions.

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
