# Peer review of "What Is the Impact of COVID-19 Pandemic on Patients with Pre-Existing Mood or Anxiety Disorder? An Observational Prospective Study"

_medicina, 2021, doi:10.3390/medicina57040304_

Round 1
Reviewer 1 Report
manuscript improved after revision.
raised issues have been reported mainly as astudy limitation by authors.
Author Response
Thank you for your constructive comments that helped us to improve the quality of our manuscript.
Reviewer 2 Report
This is an interestingpaper on psychopathological impact of
pandemic related stress in patients already
affected by anxiety disorders, OCD or mood disorders
I only have some minor concerns:
- there are some English words misspelling
in the Introduction section (‘word’
or Maj’)
- some abbreviations were not previously mentioned (e.g YNOCS)
- I wonder if the authors investigated COVID
related stress with specific questions or assessment scales.
Reviewer 3 Report
In this manuscript, the authors investigated the psychological and psychopathological impact of the pandemic stress on patients with pre-existing mood, anxiety, and obsessive-compulsive disorder (OCD). This is an important topic and has captured much attention in the scientific community.
The authors concluded that the continuity of psychiatric care protected the patient against repeals or symptom worsening. However, the manuscript does not provide enough evidence for this statement, which makes it difficult to draw any conclusion. In addition, as also mentioned by authors, the results are in contrast with previous studies that reported patients with pre-existing psychiatric disorders had a higher psychological vulnerability to impacts of the pandemic than the general population. The manuscript will benefit from further clarification and explanation.
I have a few comments/suggestions:
Title:
- “Continuity of the care protects against relapse or symptom worsening …”. There was no control group in this study to analyze a protect effects. Therefore, this title can be very misleading for the readers.
Abstract:
- The methodology is not outlined in the abstract.
Introduction
- Since the results of this manuscript are in contrast with previous studies, a more thorough summary of the previous studies is appreciated in introduction and/or discussion section.
Materials and Methods
- How did the authors measure the psychological impact of pandemic stress? Did they use a standardized scale?
Results
- The authors analyzed the psychological impact of pandemic stress. However, as it appears in the text, the patients reported their own experience of “higher or lower distress than their family and friends”. It is difficult to evaluate the reliability of this information, especially during the lockdown when the patients had less possibilities to interact with their family and friends.
- The sample size for OCD and PD was too small to conclude.
- It is written,” The number of the phone calls for psychiatric emergencies received from March 10 to 155 June 30 2020 did not differ significantly from these received from March 10 to June 30 156 2019..”. This result is rather confusing. As Figure 1 shows, the number of phone calls in 2019 increased from March to June. The author did not provide any reason for that.
- More information about the treatment would be useful to understand the results. For example, were the patients using a higher dose of benzodiazepines (or other psychotropic drugs)? Did they start a new medication to deal with their stress?
Discussion
- As mentioned above, a more carefully written summary of the previous studies is needed.
- The authors have well documented the limitation of their study. I suggested adding a paragraph about the strength of their study.
Round 2
Reviewer 3 Report
I have no further comments or remarks.
This manuscript is a resubmission of an earlier submission. The following is a list of the peer review reports and author responses from that submission.
Round 1
Reviewer 1 Report
The results did not confirm the high vulnerability to direct and indirect consequence of COVID-19 pandemic stress of patients with pre-existing AD, OCD or MD when continuity of the care provided. The continuity of the care can reduce the risk of symptoms exacerbation also in patients with OCD, more sensitive to pandemic stress. It is valuable result for the doctors and practical conclusion of the study.
Reviewer 2 Report
authors reported their clinical experience in patients with anxiety disorders. The article is very interesting and i suggest few modifications to the manuscript in order to modify it soon.
- authors should add a specific paragraph and a specific table for the possible increased consumption of drugs with action toward modification of anxiety disorders.
- - from a technical point of view Authors have not a control group and this is a great limitation for the study: i suggest them to perform a cut to the actual presentation and to focus all available data in a good short report .